# Position: Why Current Fair-AI Fails Spatial Fairness, And How to Adapt to It

**Nripsuta Ani Saxena** [1]   **Abigail L. Horn** [1]   **Wenbin Zhang** [2]   **Cyrus Shahabi** [1]

## Abstract

Despite location being increasingly used in decision-making systems deployed in sensitive domains such as mortgages and insurance, little attention has been paid to the unfairness that may seep in due to the correlation of location with characteristics considered protected under anti-discrimination law, such as race or national origin. This position paper argues for the urgent need to consider fairness with respect to location, termed *spatial fairness*. It outlines the harms perpetuated through location's correlation with protected characteristics, which may be particularly consequential due to its treatment as a neutral or purely technical attribute, abstracted from its historical, political, and socioeconomic context. This interdisciplinary work connects knowledge from fields such as public policy, economic development, and geography to highlight how existing fair-AI research falls short in addressing spatial biases, and fails to consider challenges unique to spatial data. Furthermore, we identify limitations in the small body of prior work on spatial fairness work, and propose guidelines to inform future research aimed at mitigating spatial biases in data-driven decision-making systems.

## 1. Introduction

The adoption of data-driven decision-making to learn patterns from enormous amounts of historical data has surged in the last decade in socially impactful domains such as finance, housing, and public services, reducing human effort while increasing efficiency. However, as the fairness in AI (fair-AI) literature notes (Mehrabi et al., 2021; Pessach & Shmueli, 2022), such systems can replicate–or even amplify–undesirable patterns in historical data, such as racial bias, in ways that systematically disadvantage marginalized groups.

This observation has led to research on developing fairer AI systems, centered on protecting against discrimination with respect to personal characteristics or traits protected by anti-discrimination law, commonly referred to in the literature as *(legally) protected characteristics or attributes*. However, even when this legislation is adhered to, discriminatory effects may persist because legally protected attributes can have proxies in the form of non-protected characteristics that are statistically or structurally entangled with them – such as location. Extensive research in public policy, economic development, and sociology has documented the extensive discrimination in the form of spatial discrimination that still adversely affects society today. For example, research has documented discrimination against racial minorities in the mortgage industry (Pager & Shepherd, 2008; Ross et al., 2008; Massey et al., 2016), despite the Equal Credit Opportunity Act prohibiting creditors from discriminating against applicants on the basis of race (Mehrabi et al., 2021). This is likely driven by the presence of extensive correlation of race with zip codes in the U.S. due to prejudiced practices such as redlining[1] prevalent in the past (Schill & Wachter, 1994; Woods, 2012) rather than overt prejudice.

The persistence of historical biases that disproportionately disadvantage racial and ethnic minorities (Rothstein, 2015; Schill & Wachter, 1994; Rothstein, 2017) in the outcomes of decision-making systems—despite legal protections for certain attributes—is alarming and warrants greater attention. While many attributes may be correlated with a legally protected attribute, the use of location in decision-making systems is especially consequential because it is often jointly correlated with multiple legally protected characteristics, such as race or national origin (Marshall, 1974; Willborn, 1984; Romei & Ruggieri, 2014; Becker, 2010). Table 1 lists protected characteristics inferable from location data.

Several approaches have been proposed to address the lack of consideration of legally protected attributes and their proxies in decision-making systems. Simply not considering protected attributes in decision-making, an approach

---

[1]Department of Computer Science, University of Southern California, Los Angeles, USA [2]Department of Computer Science, Florida International University, Miami, USA. Correspondence to: Nripsuta Ani Saxena <nsaxena@usc.edu>.

*Proceedings of the $43^{rd}$ International Conference on Machine Learning*, Seoul, South Korea. PMLR 306, 2026. Copyright 2026 by the author(s).

---

[1]Redlining is the discriminatory practice of systematically denying mortgages, insurance, and other financial services to residents of specific neighborhoods based on their race or ethnicity(Woods, 2012; Rothstein, 2017).

known as 'fairness through unawareness' (Grgic-Hlaca et al., 2016; Kusner et al., 2017) has been shown to be insufficient in practice because historical data often encode discriminatory patterns through correlated features and downstream outputs (Lee & Floridi, 2021). Other fair-AI work has argued to "repair" or fix attributes that may act as *proxies* for legally protected characteristics (He et al., 2020; Kamiran et al., 2010; Berk et al., 2017; Zafar et al., 2017). However, subsequent research has shown this is ineffective in practice (Corbett-Davies & Goel, 2018; Lee & Floridi, 2021). Other work argues for dropping both proxies and protected characteristics from the data altogether (Datta et al., 2017), but this is not ideal as a proxy attribute can carry other information relevant to legitimate decision-making objectives, such as crime rates associated with zip codes in mortgage underwriting. These limitations of existing approaches motivate the need for fairness definitions and methods tailored to address spatial biases. Future work will need to overcome challenges unique to spatial data to address fairness concerns and extend the generalizability of existing fair-AI methods.

This work advocates for a dedicated focus on spatial fairness within fair-AI. We use the interplay of legislation, historical discrimination, and AI systems within the U.S. to motivate greater attention towards bias due to location throughout the paper, however, the issue of discrimination due to location because of its correlation with other attributes is not limited to the U.S. alone (Galster & Friedrichs, 2015; Korsu & Wenglenski, 2010). By drawing on literature in the law, sociology, and economic development as well as computer science, we aim to ensure that future research by computer scientists in fair-AI does not fall into the trap of not being implementable in practice due to its noncompliance with legal or other standards (Xiang & Raji, 2019; Barocas & Selbst, 2016; Kim, 2022). We delineate challenges specific to location and draw upon relevant research in fields such as public policy and the law to establish guidelines for future research in spatial fairness. Our main contributions are:

- Motivate the importance of systematically addressing spatial fairness within the context of fair-AI, drawing upon scholarly literature from the law and public policy
- Clearly outline challenges unique to spatial data
- Identify limitations of the handful of spatial fairness definitions and techniques proposed in the computer science literature so far and why they are insufficient
- Draw from fields of public policy, law, transport geography and economic development to delineate guidelines for future research on spatial fairness in fair-AI

Paper structure. Section 2 unpacks spatial bias, its societal impact, and the emerging legal landscape on zip-code discrimination. Sections 3 and 4 identify challenges unique to spatial data that limit fair-AI generalizability and limitations of current scholarship. We conclude with guidelines for future work in Section 5 and alternative views in Section 6.

## 2. Context and Motivation: Location bias

We discuss spatial segregation in the U.S. and its lasting effects, and outline the case for legal protections against location-based discrimination, especially zipcodes.

### 2.1. Spatial unfairness in the real world

Although the fair-AI literature has yet to explore bias due to location in depth, it has been extensively covered in fields such as public policy, urban planning, and the law (Schill & Wachter, 1994; Rothstein, 2017; Baum et al., 2015). For example, neighborhoods in the U.S. have been historically correlated with race, a legally protected characteristic in many domains (e.g. housing, employment, credit), in complex ways due to prejudiced practices such as redlining. Despite being prohibited by the Fair Housing Act in 1968 [42 U.S.C. §§ 3601 - 3619], effects of redlining continue to linger (Angwin et al., 2017; Mehrabi et al., 2021). Zip codes in many parts of the country still correlate with race, potentially acting as a proxy for race in data-driven decision-making scenarios where zip codes are considered as a feature, such as in mortgages and insurance. This is well documented. For example, the non-profit ProPublica showed that drivers who live in minority neighborhoods are charged higher insurance premiums than drivers of same risk levels who reside in predominantly white neighborhoods (Angwin et al., 2017). Research has also found prices for ride-hailing trips in the city of Chicago to or from neighborhoods with larger non-white populations or higher poverty levels were significantly likely to be charged higher prices (Pandey & Caliskan, 2021; Saxena et al., 2023). Spatial bias has also been observed in commercial logistics algorithms. A Bloomberg study found Amazon Prime's Same-Day Delivery service disproportionately excluded minority neighborhoods, and these spatial disparities could not be explained by income differences alone (Ingold & Soper, 2016).

Neighborhoods (and their zip codes) may also be correlated with immigrants, i.e., regions where immigrants are more likely to live, which can act as a proxy for national origin, another protected characteristic in various domains under U.S. law (Mehrabi et al., 2021). Moreover, neighborhoods may also be correlated with income (Mode et al., 2016). Beyond the ride-hailing disparities noted above, low-income neighborhood residents may also face other forms of income-related discrimination. For example, families receiving housing vouchers often face barriers in moving to lower-poverty and higher-income neighborhoods from landlords who discriminate against their source of income (Mazzara & Knudsen, 2019). As a result, many families receiving housing vouchers still end up in low-income, racially segregated neighborhoods despite being able to afford homes in better areas (Rothstein, 2015). Such families might suffer doubly if they are also then subjected to spatial discrimination due to

| Attribute | Protected at level | Correlation w/ location | References & Notes |
|---|---|---|---|
| Race | Federal | ✔ | (Rothstein, 2015; 2017; Fulwood III, 2017) |
| National origin | Federal | ✔ | (Pandey & Caliskan, 2021; Ruther et al., 2018) |
| Source of income | State | ✔ | (Mazzara & Knudsen, 2019) Protected in housing in 15 states (pol) |
| Gender | Federal | ✔ | (Zhong et al., 2015; Riederer et al., 2016) |

*Table 1.* Protected characteristics in housing (federal: Fair Housing Act; state: Policy Surveillance Program), credit (Equal Credit Opportunity Act, and employment (Equal Employment Opportunity Commission. (Mehrabi et al., 2021; Chen et al., 2019; pol)

their low-income, racially segregated neighborhood. While not federally prohibited, several states have laws to protect from discrimination based on source of income (Table 1).

Despite racial segregation in housing being banned in the U.S. in 1968, spatial segregation by race has persisted, resulting in "racially homogenous public institutions that are geographically defined," (Fulwood III, 2017). As a consequence, such neighborhoods have reduced access to economic, educational, and social opportunities (Chetty et al., 2016; Li et al., 2013), which further hinders future prospects for residents in those neighborhoods. This also manifests in house prices, with a study finding that the racial composition of a neighborhood was a stronger determinant of a home's appraised value in 2015 than in 1980 (Howell & Korver-Glenn, 2021). Furthermore, research has shown that prices for houses in majority-white neighborhoods are significantly higher than for similar houses in neighborhoods that happen to be majority-black (Rothwell & Perry, 2022), worsening the wealth gap between whites and racial minorities, especially African Americans, since a home is typically a family's most significant financial asset according to a report by Pew Research Center (Taylor et al., 2011). There has been no significant change over the past few decades in the manner in which majority-black neighborhoods continue to be encircled by spatial disadvantage (Sharkey, 2014). Ethnographic research has documented that even middle class African-American families with an annual income of at least $100,000 tend to live in neighborhoods with the same disadvantages as the average white family with an annual income of $30,000 or lower (Sharkey, 2014).

Due to such extensive correlation of spatial data with legally protected attributes, decision-making systems that have been developed to be fair with respect to legally protected characteristics directly may still exhibit unfair behavior.

While our analysis leverages U.S. anti-discrimination law to motivate this problem, spatial inequities are a global issue. Research economics and public policy has documented rising spatial inequities in countries across the world (Kanbur & Venables, 2005; Kochendörfer-Lucius & Pleskovic, 2009; Zhao & Tong, 2000; Feldman & Storper, 2018).

## 2.2. Legal support for spatial fairness

U.S. law recognizes two forms of discrimination against protected classes (Lee & Floridi, 2021; Baum et al., 2015): disparate treatment and disparate impact. In layman terms, disparate treatment refers to direct or intentional discrimination while disparate impact refers to indirect or unintentional discrimination (Barocas & Selbst, 2016; Primus, 2010). While disparate treatment focuses on the *intent* to discriminate, disparate impact focuses on *outcomes* that are discriminatory towards members of a protected class despite a seemingly neutral decision-making policy.

The law offers protections against both forms of discrimination on the basis of legally protected characteristics in domains such as housing [42 U.S.C. § 3601 et seq], credit [15 U.S.C. 1691 et seq.], and employment [42 U.S.C. § 2000e]. Some states define additional protected attributes: marital status is also a protected class in the state of California in the domain of employment (Gelb & Frankfurt, 1982) although it is a protected characteristic only in the domain of credit at the federal level. Typically under U.S. anti-discrimination law, protected characteristics are those which are considered *immutable*, i.e., characteristics that were not chosen and cannot be changed over time (Hoffman, 2010), such as race, color, and sex. Race and color were among the first classes to be considered immutable in the eyes of the law, and deemed protected classes. In the recent past, however, this definition has been expanded in a series of court judgements to include characteristics that are deemed difficult or too significant to ask someone to change. For instance, the U.S. Supreme Court issued a landmark judgment in June 2020 that found that discriminating against a person on the basis of their sexual orientation or gender identity in employment violates the protection for discrimination under sex in Title VII of the Civil Rights Act of 1964 [Bostock v. Clayton Cty., 140 S. Ct. 1731 (2020)].

We argue that residential zip codes can also be considered an immutable characteristic for a section of the population, thereby deserving protection against discrimination on the basis of zip codes. Common sense dictates that it is not easy or straightforward for an individual or family to move to a

lower-poverty or less segregated neighborhood in order to access better economic or educational opportunities. Even when disadvantaged families can afford to live in higher-income, more integrated neighborhoods with the support of housing vouchers they still cannot move due to source of income discrimination by landlords (Mazzara & Knudsen, 2019). Research has also shown that such discrimination overwhelmingly affects minorities, perpetuating systemic racism "and denies renters of color, especially Black renters, equal access to housing opportunities." (Coa, 2023) Therefore, for low socioeconomic status families, where they live can be considered an immutable characteristic – they simply do not have the ability to move even if they wish or can afford to. Furthermore, they are forced to remain in their low-income neighborhoods which are likely to be correlated with race due to historical segregation, suffering the negative consequences associated with that in addition to the discrimination due to their source of income.

Due to the amalgamation of these negative factors, the groups affected most severely are people of color, people with disability, and women, which may be considered disparate impact. This line of thinking is gaining traction. After ProPublica published analyses documenting higher insurance premiums for drivers living in minority neighborhoods than similar-risk drivers in majority-white neighborhoods (Angwin et al., 2017), former Illinois State Senator Jacqueline Collins proposed a bill banning the use of zip codes in determining insurance quotes as they can act as proxies for race (Salcedo, 2017). While that did not pass, Illinois State Representative Will Guzzardi introduced the House Bill 2203 in February 2023 that proposes to end the use of non-driving factors including zip codes in setting insurance rates due to their well-known correlation with race.

The tangible impact on society of location's correlation with protected attributes is undeniable. Regardless of legal protections given to location, the AI community has an ethical responsibility to strive to address it given the extensive societal impacts of our systems. Doing so requires confronting challenges specific to spatial data, as we shall see next.

## 3. Problems unique to spatial data

Standard fair-AI methods ignore the unique challenges of spatial data, which we detail below.

### 3.1. Dimensionality

A critical distinction between spatial fairness and typical fair-AI work lies in the cardinality of the protected attribute. Fair-AI research considers attributes like race or gender, which possess a limited number of distinct values. However, when location is treated as a protected attribute, the number of values can expand to thousands (e.g., zip codes).

This high cardinality presents a fundamental barrier for existing dimensionality reduction techniques. For instance, Samadi et al. (2018) propose Fair-PCA, but their analysis shows that if a protected attribute has $k$ possible values, the algorithm requires a target subspace of $k + 1$ dimensions to satisfy fairness constraints. While this is feasible for binary attributes ($k = 2$), it is intractable for spatial data. With thousands of location values, the required dimensionality would effectively negate the purpose of reduction. Consequently, generic fair dimensionality reduction methods are ill-suited for spatial scenarios where the protected attribute is defined by fine-grained geographic units.

Fair dimensionality reduction is non-trivial and spatial data's correlations with many protected characteristics (e.g., race, ethnicity, national origin), with each characteristic having multiple subgroups, complicates fair dimensionality reduction in spatial decision-making even more. Techniques in other areas such as r-trees (Guttman, 1984) that are designed to take advantage of the unique relationships between the two spatial dimensions to improve performance for multi-dimensional spatial data can offer insights for leveraging unique spatial dynamics to improve upon generic dimensionality reduction methods for spatial scenarios.

### 3.2. Complexity of computing spatial network distance

Computing distance between two locations involves crucial design choices with practical implications. The commonly used Euclidean distance, i.e., the distance between points $A$ and $B$ in a straight line, is straightforward to compute but does not reflect road network distance, i.e., the distance between $A$ and $B$ by road. More complex to compute, road network distance often differs from Euclidean distance in practice (Sander et al., 2010; Apparicio et al., 2003). Moreover, in certain decision-making scenarios considering travel time, i.e., the time it takes to travel between $A$ and $B$, is crucial. For example, houses equidistant to a hospital in road distance may still have very different reachability to the hospital in practice due to traffic patterns and/or lack of public transport making the travel time for houses in one zip code much worse than another (Anastasiou et al., 2019). The time-dependent, dynamic nature of traffic adds further complexity to the problem (Li et al., 2017). Furthermore, as distance metrics are central to individual fairness, increased complexity due to road network distance and travel time increases difficulty for individual spatial fairness as well.

### 3.3. Continuity of space

Geographic spaces are continuous in nature, and not as easy to analyze as discrete spaces (Xie et al., 2022). Discretizing spatial data is not straightforward due to features of spatial data such as the spatial structure, association, and heterogeneity (Cao et al., 2014; Haining, 2003). Moreover,

geographical partitionings are heavily context-dependent, and less likely to have partitionings that are universally well-accepted than other continuous variables such as income which has well-defined thresholds (e.g., for income tax). Additionally, when the decision-making scenario is spatio-temporal, e.g., involves travel time, there is one more continuous dimension – time, further complicating matters. Finally, achieving fairness with respect to continuous attributes is non-trivial. Most fair-AI work focuses on discrete variables (e.g., gender, race) (Mehrabi et al., 2021). Although recent work explores fairness with continuous attributes, many depend on computationally intractable statistical independence measures in practice (Jiang et al., 2022; Mary et al., 2019; Jha et al., 2021; Creager et al., 2019), lack estimation accuracy guarantees (Jiang et al., 2022; Cho et al., 2020; Roh et al., 2020), or are not sensitive to protected subgroups less likely to occur in the population (Jiang et al., 2022). The combined challenges of continuous spatial data in two or three dimensions (with time) and fairness with continuous attributes make it a nontrivial task.

### 3.4. Modifiable Areal Unit Problem (MAUP)

Identified in 1934 (Gehlke & Biehl, 1934) and coined as MAUP in 1979 (Openshaw, 1979), it is a type of statistical bias observed in geography. Conceptually, MAUP can be considered a spatial analogue of Simpson's Paradox. At a high level, two types of biases can distort results when analyzing spatial data. The scale effect occurs when point data is aggregated at different levels of aggregation. Despite using the same point data, different aggregation levels can represent different patterns. In contrast, the zonal effect occurs when spatial is grouped by different artificial boundaries, leading to different results. In other words, data collected for the same region but at different spatial scales will give inconsistent results. Research has found MAUP can be highly unpredictable in multivariate statistical analyses, with effects that are often substantially more severe than those observed in univariate or bivariate analyses(Fotheringham & Wong, 1991). This fragility is well documented across disciplines: research shows the same data can yield conflicting conclusions about residential segregation depending on the spatial unit chosen (Jones et al., 2018). MAUP also affects ecological model inferences in disease mapping due to substantial variation in both scale and zoning effects (Hogg et al., 2025). Another common real-world example of MAUP is gerrymandering.

Concrete, real-world examples illustrate the tangible harms of ignoring the correlation between location and protected attributes. For instance, a ProPublica investigation found that auto insurers charged drivers in minority neighborhoods premiums up to 30% higher than drivers of equivalent risk residing in predominantly white areas (Angwin et al., 2017). This discovery prompted California regulators to mandate

corrective measures (Angwin & Larson, 2017). Similarly, Bloomberg revealed that Amazon Prime's same-day delivery service disproportionately excluded majority-Black neighborhoods, a disparity that income differences could not justify. After pressure from members of Congress and the mayors of New York City and Boston, Amazon committed to fill all racial gaps in the service's coverage (Sottek, 2016). In both instances, the bias was spatial: individuals were treated differently based on where they lived because geography acted as a proxy for race, and identifying this spatial pattern was the essential first step towards corrective action and regulatory accountability.

### 3.5. Spatial autocorrelation

> "Everything is related to everything else, but near things are more related than distant things."
> – Waldo Tobler, the First Law of Geography

Finally, there is the challenge of spatial autocorrelation: nearby things are more related than distant ones. This violates the independence assumption in many standard statistical tests, requiring careful selection of statistical methods. Recent works in ML study such spatial dependence in different fields. Deppner et al. (2022) propose a spatial cross-validation strategy to reduce spatial bias in tree-based algorithms commonly used for price and rent modeling in the real estate industry. In the geosciences, work explores spatial autocorrelation when spatial features like spatial lag are used in random forests (Liu et al., 2022), and incorporating spatial autocorrelation measures in ML modelling to predict geotechnical information like ground surface elevation (Kim et al., 2023), or increasing the prediction accuracy of heavy metals in soil (Sergeev et al., 2019). However, none of them look at spatial correlation with respect to legally protected attributes. Majumdar et al. (2022) propose a hypothesis testing framework to detect spatial autocorrelation and its strength in presence of correlation with a protected attribute, although they do not propose techniques to mitigate the observed bias. Thus, despite significant practical applications, mitigating spatial autocorrelation and its interplay with protected attributes remains deeply understudied.

These complexities hinder use of fair-AI work in spatial settings, motivating dedicated spatial fairness research.

## 4. Limitations of current spatial fairness work

However, early work on spatial fairness definitions and techniques fall short in significant ways, as we detail below.

### 4.1. Spatial fairness metrics: Legal, technical soundness

Many fair-AI fairness definitions will not withstand legal scrutiny (Xiang, 2020; Wachter et al., 2021), making them inapplicable in practice. Worryingly, they may be

incompatible with U.S. anti-discrimination and E.U. non-discrimination law. Incompatibility with anti-discrimination law may also be an issue for the handful of spatial fairness definitions proposed so far. Shaham et al. 2022 propose two definitions: distance-based and zone-based spatial fairness. The former focuses on decision-making scenarios based on how far an individual is from a reference point, such as a shop offering discount vouchers to customers within a certain distance. The latter considers decisions based on whether an individual falls within particular geographic zones. However, both definitions only consider location while evaluating fairness, without accounting for legally protected attributes. This substantially limits their real-world applicability, since making decisions based on location alone is not unlawful. Related work (Sacharidis et al., 2023; Xie et al., 2022) similarly considers fairness only with respect to location, limiting their practical usefulness. To ensure adoption in the real-world, it is imperative that future spatial fairness work is aligned with established legal frameworks for anti-discrimination.

Instead of replacing existing fairness definitions, spatial structure introduces an additional layer that interacts with them. While classical fairness notions are defined over the outcome space, spatial dependencies introduce correlations across individuals or regions, which can lead to systematic disparities that are not captured by standard metrics. We examine how spatial data can undermine two broad families of fairness metrics in distinct ways.

*Group fairness metrics.* Standard metrics like demographic parity or equalized odds evaluate outcomes across subgroups of a protected attribute (e.g., race: white/black). When location is treated as a protected attribute, issues arise because they weren't designed for the messiness of spatial data, like high cardinality (having thousands of zip codes as groups), which makes reliable estimation of outcome rates per group difficult for metrics like demographic parity and calibration. MAUP also has an effect, e.g., model may satisfy demographic parity under one administrative partition (e.g., census tracts) but violate it under another (zip codes).

Beyond boundaries, statistical assumptions of these metrics often fail in a spatial context. For example, equalized odds relies on the independence of data points, but Tobler's First Law of Geography (nearby things are more related) means that observations are naturally correlated. Spatial autocorrelation violates the core statistical assumption of independence of tests used to evaluate bias.

*Individual fairness metrics.* These metrics require similar individuals to receive similar outcomes, with similarity defined by a task-specific distance metric. Prior spatial fairness work has operationalized this by defining individual similarity as a function of the distance b/w two individuals' locations, without accounting for other attributes. This is

legally insufficient since location is not a legally protected attribute in isolation. The true ethical concern arises when geography acts as a proxy for protected attributes like race.

We propose that the path forward consists of joint metric frameworks. Instead of treating location and protected attributes independently, standard metrics can be extended by aggregating them over spatial neighborhoods, or by introducing spatial regularization terms that enforce fairness across similar regions. This would be consistent with the underlying legal and ethical concerns, while allowing practitioners to retain familiar fairness criteria and account for spatial effects that are otherwise overlooked.

**4.2. Spatial fairness techniques: limitations**

4.2.1. DO NOT DEMONSTRATE REDUCTION IN BIAS

A critical limitation of current spatial fairness techniques is the failure to "close the loop." While these methods are motivated by location's correlation with protected characteristics, they rarely validate that their spatial interventions actually achieve this goal (Shaham et al., 2022; Sacharidis et al., 2023): they lack experiments showing that addressing location-based bias successfully mitigates unfairness toward the underlying protected groups. To be considered for adoption, techniques must unequivocally demonstrate they result in tangible improvements for the protected characteristics.

4.2.2. TAKE AGENCY AWAY FROM PEOPLE

Some spatial fairness work proposed so far uses a fuzzy boundary instead of hard boundaries as are typically employed in domains such as education (e.g., school districts) or housing. While this may have some technical advantage, a significant consequence of this would be to take agency away from people. For example, with clear school-district boundaries, a low-income family with school-age children can choose to make sacrifices to live in a better school district (Hristov & Khare, 2016). If such boundaries are fuzzy, however, then this can impact the decision-making ability of families in a deeply negative manner. Wealthy families will still be able to ensure a good education for their children by living squarely in the heart of good school districts or simply paying for private schools. But lower income families will suffer due to the opaqueness of the fuzzy boundary, and not being able to assess the chance that a particular address may have of a good public school. Past research shows that a big barrier to disadvantaged families benefiting from public benefits they are eligible for is lack of knowledge of eligibility rules (Wu & Eamon, 2010; Anderson, 2002; Stuber & Kronebusch, 2004; Zedlewski, 2003) and complicated application processes (Wu & Eamon, 2010; Morrow & Horner, 2006; Zedlewski et al., 2002). Therefore, fuzzy boundaries may disproportionately harm those who are least equipped to navigate the process efficiently.

### 4.2.3. DO NOT BUILD UPON EXISTING KNOWLEDGE

Research is about building on existing work to stand on the shoulders of giants. Literature in spatial scan statistics (Wong, 2004; Fotheringham & Wong, 1991) and transport geography (Guo & Bhat, 2004; Mitra & Buliung, 2012) exploring MAUP from different angles can offer ideas for how to grapple with it in spatial fairness. Similarly, work in public policy studies spatial segregation patterns with respect to race, ethnicity, and immigration in housing and schools across the country (Schill & Wachter, 1994; Sanchez, 2006; Saiz & Wachter, 2011; Schill & Wachter, 1995). Economic research can offer guidance on which spatial measures help improve economic development for the disadvantaged (Chetty et al., 2016; Brata et al., 2009; Glasmeier, 2018).

## 5. Guidelines and Future Directions

We mitigate spatial bias by rethinking the modeling pipeline and establishing guidelines from foundation to deployment.

### 5.1. Foundational Steps

#### 5.1.1. ESTABLISH SOCIO-TECHNICAL CONTEXT

**Clearly define domain-specific constraints and requirements.** Before technical development, researchers must document the legal, regulatory, and public policy constraints governing decision-making systems in that domain. In other words, document the constraints or requirements the system is expected to satisfy.In addition, identify the population(s) likely to be impacted by the decision-making system, and the subgroups most vulnerable to marginalization.

#### 5.1.2. DATA COVERAGE AND REPRESENTATIVENESS

**Evaluate training data for gaps in coverage and demographic representativeness.** Location data collection is often systematically biased, yielding samples that reflect only a subset of the underlying population due to skewed data sources and collection practices. For instance, location data collected from mobile applications may underrepresent certain demographics, as device type (e.g., iPhone vs android) and app usage patterns vary significantly by socioeconomic status and age (Zeighami & Shahabi, 2024). Researchers should explicitly document these gaps and any reweighting or other procedures used to mitigate them.

#### 5.1.3. ENGAGE LEGAL, COMMUNITY STAKEHOLDERS

**Collaborate with experts and affected communities.** Identify stakeholders, including experts who shape, interpret, or enforce the legal, regulatory, and public policy constraints in the decision-making domain. They can aid computer scientists translate the normative requirements into precise technical specifications and evaluation criteria, enabling verification that system outputs meaningfully satisfy the constraints. Moreover, in high-impact domains like housing, community stakeholders, especially those underrepresented in the data, should be consulted be consulted during development and evaluation. This sustained engagement can uncover factors and values important to the general public that may not be captured in formal requirements, like distinct school district boundaries so communities retain meaningful agency in how decisions affect them.

### 5.2. Model Design Principles for Spatial Fairness

#### 5.2.1. TAILORING FAIRNESS TO DOMAIN AND THE LAW

**Develop domain-specific fairness definitions.** Spatial fairness definitions must be tailored to the specific problem domain in collaboration with stakeholders identified in the foundational phase. Legal experts are particularly vital to ensure the proposed definition: (1) strictly complies with the domain-specific anti-discrimination laws; (2) prevents the inadvertent encoding of discriminatory intent (i.e., disparate treatment); and (3) enables evaluations to meaningfully assess whether system outputs avoid disparate impact.

Moreover, extensive debate in the fair-AI community has established that no single fairness definition is appropriate for all decision-making contexts (Saxena et al., 2019; Mehrabi et al., 2021). The same is likely true for spatial fairness, given that decision-making objectives and acceptable trade-offs vary widely across domains. For example, spatial decisions in the public sector are often driven by equity-oriented goals rather than profit, such as an equitable assignment of students to public schools. In contrast, private sector decision-making typically involves profit incentives alongside fairness objectives, while regulated industries such as mortgage lending and insurance must also satisfy specific federal and state anti-discrimination requirements.

Consequently, spatial fairness definitions should clearly specify the intended domain, industry context, and the legal and policy constraints they aim to satisfy. This enables meaningful compliance evaluation and delimits the scope of generalizability, recognizing that a definition suitable for one spatial setting may not transfer to others. Explicitly articulating these assumptions establishes a foundation for deployment and rigorous, comparable future research.

#### 5.2.2. A JOINT PROTECTED ATTRIBUTE FRAMEWORK

**Treat location as a protected or immutable characteristic.** Immutability is deeply associated with anti-discrimination law in the U.S. (Hoffman, 2010). Recent interpretations are a characteristic "that either is beyond the power of an individual to change," [In re Acosta, 19 I. & N. Dec. 211, 233-34 (B.I.A. 1985)] or in other terms, "either unchangeable in absolute terms or so fundamental to identity or conscience that

individuals effectively cannot and should not be required to change it" [In re Acosta, 19 I. & N. Dec. 211, 233-34 (B.I.A. 1985)] or "changing it would require great physical difficulty, such as requiring a major physical change or a traumatic change of identity." [Watkins v. U.S. Army, 875 F.2d 699, 726 (9th Cir. 1989)] (Hoffman, 2010), thereby establishing a characteristic need not be unchangeable to be considered immutable under anti-discrimination law. This, in addition to ongoing efforts to afford protection against discrimination on the basis of zip codes, brings us to the next guideline: consider location a protected attribute along with traditional legally protected attributes like race, or at least an immutable one. We outline example use cases below.

In domains with a demonstrated history of location-related bias, zip codes could be considered a joint protected attribute with the legally protected attributes it perpetuates bias against. An intuitive measure of individual fairness is to treat similar people similarly (Dwork et al., 2012). With location and race as a joint protected attribute, two applicants, $i$ and $j$, who are similar on non-protected aspects should receive similar decisions, $D(i) \approx D(j)$, regardless of race and location. When impractical, location should be treated as an immutable attribute. For example, some fair- and explainable-AI research give applicants actionable points to increase their likelihood of a positive decision (Ustun et al., 2019). Suggested actions should, however, be reasonably achievable. For example, a model may recommend increasing annual salary by $2,000 to improve loan approval odds, but if the salary of an applicant with a PhD degree is low for someone with that degree, the model cannot recommend they undo their PhD. Similarly, residence should be treated as immutable for low-income families, i.e., something they cannot reasonably change since they often do not have the freedom to make this choice, sometimes even if they can afford to (source of income discrimination, see Section 2.1).

### 5.2.3. DISRUPT FEEDBACK LOOPS VIA EXPLORATION

**Incentivize exploration to break cycles of neglect.** To prevent the entrenchment of historical biases, models should encourage an exploration-exploitation trade-off. By incentivizing collection of fresh data from under-sampled regions rather than solely optimizing for known rewards, systems can disrupt feedback loops where marginalized neighborhoods are systematically neglected or unfairly targeted based on a prejudiced past. This approach draws on research into feedback loops in predictive policing models (Ensign et al., 2018), and work exploring how to break such cycles (Joseph et al., 2016; Jabbari et al., 2017).

### 5.2.4. MITIGATING MAUP RESISTANCE

**Ensure robustness to the Modifiable Areal Unit Problem (MAUP).** Since results in spatial analyses can vary sub-stantially depending on how geographic units are defined, spatially fair systems should be robust to MAUP. MAUP manifests through two primary mechanisms: the zonal effect, in which results change when geographic boundaries are redrawn, and the scale effect, in which outcomes shift as data are aggregated to coarser or finer spatial resolutions. These sensitivities can undermine fairness claims. For example, a system may appear equitable under one administrative partition (e.g., census tracts) but exhibit disparate impact under another (e.g., zip codes), or fairness assessments may change simply due to arbitrary aggregation choices.

To mitigate these risks, spatial fairness research should incorporate robustness checks across multiple (plausible) zoning schemes and spatial scales, and explicitly report how fairness metrics vary under alternative unit definitions. Related work in spatial statistics and geography offers ideas for mitigating MAUP in applied settings, such as traffic safety (Xu et al., 2018) and spatial public health analyses (Tuson et al., 2019; Tuson, 2022; Hennerdal & Nielsen, 2017). Other work explores the design of zone-based districts for constructing partitions that align with domain objectives while avoiding MAUP-driven artifacts (Martin, 2003; Tuson, 2022). Spatial fairness research can build on these foundations and domain expertise to determine the appropriate spatial resolution and partitioning scheme for the problem at hand, and ensure fairness evaluations remain well-founded when assessed under plausible alternative representations.

### 5.3. Validation and Monitoring

#### 5.3.1. EMPIRICALLY VERIFYING BIAS MITIGATION

**Explicitly document bias reduction against protected attributes.** Although spatial fairness work is motivated by location's correlation with protected attributes, recent works rarely demonstrate that their methods actually mitigate this bias (Shaham et al., 2022; Sacharidis et al., 2023). To ensure real-world utility, future work must explicitly evaluate spatial bias with respect to protected attributes: comparing bias levels before and after applying their methodology, while also reporting trade-offs with other objectives (e.g., prediction accuracy). For example, if a bias mitigation method transforms spatial data $L(x)$, where $x$ is a protected attribute, into $L'(x)$, it must quantify how much this actually alleviates bias against the protected attribute rather than simply assuming the spatial correction is sufficient.

We propose a three-step validation protocol to ensure spatial fairness is a robust, verifiable model property: *Spatial Proxy Analysis* to confirm spatial bias reduction against legally protected attributes by proposed methods; *Multi-Scale Auditing* to ensure fairness metrics remain stable across geographic boundaries (e.g., zip codes), guarding against MAUP; and *Dependency Clustering* with statistics like Moran's I to surface localized disparities that global averages obscure.

### 5.3.2. Continuous Auditing and Monitoring

**Establish ongoing compliance and monitoring pipelines.** Models deployed in sensitive or high-impact domains should be accompanied by an ongoing compliance and monitoring pipeline. Since demographic and spatial patterns can change over time (e.g., through gentrification, migration, or infrastructure investments), models that were initially evaluated as spatially fair may later exhibit different disparities. Regular re-auditing is therefore essential to ensure continued alignment with legal and policy requirements, and to detect emerging spatial biases before they can transform into harmful outcomes. Where practically plausible, monitoring should include periodic disparate impact assessments with clearly defined triggers for human review.

## 6. Alternative Views

**Spatial inequity is outside the scope of algorithm design.** One view is that spatial inequities are fundamentally structural, and therefore should be addressed through policy and institutional reform, rather than technical modifications to models. Critics might argue that "fixing" spatial bias in a mortgage or ride-hailing model is a band-aid that distracts from the root cause. We agree that algorithmic interventions are not a substitute for policy reform. However, data-driven systems are no longer passive observers of the city, they actively reshape it. Decision-making systems that optimize for short-term efficiency on historically biased data can entrench and accelerate spatial disadvantage, effectively "automating" the legacy of practices like redlining. Technical spatial fairness is therefore a necessary containment mechanism, a guardrail to prevents AI from amplifying structural inequities while longer-term policy reforms take hold.

While spatial inequities often originate from historical policy decisions, policymakers already rely on computational methods to address them. For example, research in public policy (Meier & Mitchell, 2022; Knaap, 2017), public health (Wang, 2020; Raza et al., 2024), and urban planning (Aaronson et al., 2021; Moro et al., 2021; Xu et al., 2025) already deploys GIS, spatial statistics, and ML to study (Aaronson et al., 2021; Meier & Mitchell, 2022; Moro et al., 2021; Xu et al., 2025; Raza et al., 2024) and mitigate them (Wang, 2020; Knaap, 2017). As the public policy community already uses them in addressing spatial inequities in various domains, excluding computational methods from the conversation would be a false dichotomy.

**Existing fair-AI approaches are sufficient.** Another view is that spatial bias does not warrant dedicated attention because it can be handled with existing fair-AI approaches. We argue that this ignores the unique statistical properties of spatial data that undermine standard assumptions in much fair-AI work. As detailed in Section 3, traditional fair-AI is insufficient because: (1) Spatial autocorrelation violates the independence (i.i.d.) assumption on which most such approaches rely. Research shows ignoring spatial autocorrelation when applying fair-AI methods actively distorts bias detection (Majumdar et al., 2022), which can cause them to produce unreliable fairness assessments. (2) Fair-AI metrics are brittle to MAUP: conducting the same analysis with data aggregated to different spatial units can yield very different conclusions (Lee et al., 2025; Manley, 2021), and existing fair-AI work is not robust to MAUP (Sacharidis et al., 2023). A real world example, gerrymandering, shows how tricky it can be to deal with it successfully (Ratliff et al., 2025).

**The efficiency-vs-fairness argument.** Practitioners in high-stakes domains may argue that imposing spatial fairness constraints will degrade system utility, increasing costs or response latencies, thereby harming the very populations the system aims to serve. We acknowledge there is often a short-term trade-off between fairness and optimization. However, ignoring spatial fairness incurs significant long-term costs. Spatially biased systems risk regulatory backlash (e.g., the legislative proposals in Illinois) and loss of user trust. Moreover, incorporating exploration into spatial models can disrupt feedback loops, potentially discovering untapped demand in underserved neighborhoods and leading to a more robust, globally optimal system rather than one stuck in a local optimum based on historical prejudice.

Finally, while optimizing for accuracy or operational efficiency can often lead to lower fairness and vice versa, this has not prevented fair-AI approaches since: (1) Many domains have a legal mandate to mitigate bias against sensitive groups regardless of the resulting cost to accuracy or efficiency (Barocas & Selbst, 2016). (2) Even in high-stakes domains with no mandates, such as healthcare or criminal justice, there is broad ethical consensus that equitable outcomes should be prioritized over marginal accuracy or gains in system performance (Chen et al., 2021). (3) Empirical evidence suggests this trade-off is often smaller than expected: well-designed fairness interventions can substantially reduce disparities with negligible loss in accuracy (Rodolfa et al., 2021). These arguments apply equally to spatial fairness.

## 7. Conclusion

This work argues for the importance of spatial fairness. We delineate the need to address persisting bias in spatial decision-making contexts and legal efforts proposed to address it, and outline problems unique to spatial analysis that compound the difficulties in addressing spatial bias computationally. Moreover, we identify drawbacks of the handful of spatial fairness work proposed so far to highlight why they fall short. Finally, we establish a set of guidelines to support future spatial fairness research avoid similar pitfalls and achieve maximal impact with adoption in the real world.

## Acknowledgments

This research has been funded in part by the NSF grant 2427150 (PI: Horn). Any opinions, findings, conclusions or recommendations expressed in this material are those of the author(s) and do not necessarily reflect the views of any of the sponsors such as the NSF.

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
