# OpenReview forum: "Position: Why Current Fair-AI Fails Spatial Fairness, And How to Adapt to It"
_ICML.cc/2026/Position_Paper_Track — ICML 2026 Position Paper Track regular_

### Official Review · Reviewer_XP7r · 2026-03-06

**Significance:** 4
**Argument Clarity:** 4
**Rating:** 5
**Confidence:** 4

**Questions:**

There are three important questions which are related to the suggestions for improvement. I will illustrate more here and hope it helps authors improve this paper.

1. Empirical Illustration of Spatial Effects

The paper argues that ignoring spatial structure can lead to misleading conclusions about fairness. Could the authors provide concrete empirical examples showing how fairness metrics change when spatial structure is properly accounted for versus when it is ignored? For example, does fairness evaluation change when different spatial resolutions (e.g., census tract vs. county vs. ZIP code) are used? How sensitive are fairness conclusions to spatial aggregation effects (MAUP)?

2. Practical Evaluation Protocol

The paper recommends accounting for spatial structure in fairness analysis, but the implementation details remain somewhat abstract. Could the authors propose a minimal protocol for evaluating spatial fairness that practitioners can follow? For instance, should such a protocol include steps such as identifying spatial proxies for protected attributes, testing sensitivity to different spatial partitions and so on?

3. Relationship with Existing Fairness Metrics

The author argues that spatial structure introduces additional complexities for fairness analysis. How should spatial fairness considerations interact with existing fairness definitions, such as demographic parity, equalized odds, or calibration? Some further discussion or exploration will help.

**Alternative Views Section:**

Yes

**Compliance With Llm Reviewing Policy A Conservative:**

Affirmed.

**Discussion Potential:**

4

**Final Justification:**

na

**Paper Summary:**

This paper states that fairness analysis in machine learning that uses geographic data should account for spatial structure. The authors claim that spatial variables often act as proxies for protected attributes due to historical and socioeconomic patterns. However, those fairness evaluations that ignore spatial structure relationships can lead to misleading conclusions and may reproduce existing inequalities.

The authors identify key pitfalls in the current most common practice, including spatial autocorrelation, aggregation effects, and the use of geographic proxies that can either mask or amplify disparities. To address these issues, they propose that fairness analysis in machine learning should incorporate spatial structural context and draw on insights from spatial statistics and related disciplines. The paper advocates developing evaluation frameworks that explicitly account for geographic/spatial structure when assessing fairness in machine learning.

**Position:**

Yes

**Position In Title:**

No

**Related Work:**

4

**Strengths And Weaknesses:**

The main strength is that the paper identifies a highly relevant problem: in the ML fairness community, spatial variables are often treated as ordinary predictive features even though they can encode protected characteristics through segregation, historical disadvantage, and geographic structure.

There are som other strengths such as clear and well-motivated position, interdisciplinary perspective, strong discussion potential, and constructive recommendations. In particular, the paper articulates a clear claim that fairness analysis must account for spatial structure, which is a realistic concern in many ML deployment contexts. It effectively connects machine learning fairness with insights from spatial statistics, geography, and policy research. The topic is likely to generate constructive discussion within the ICML community, particularly among researchers working on fairness, geospatial ML, and policy-oriented applications. The authors also propose practical guidelines for improving fairness evaluation in spatial contexts.

There are three major weaknesses. The first is on limited empirical illustration. The argument is largely conceptual and would benefit from more concrete empirical demonstrations showing how ignoring spatial structure leads to misleading fairness conclusions. The seond in on implementation. While the paper provides useful principles, it is less explicit about how practitioners should implement spatial fairness evaluation in practice. The third one is on the scope of the applicability. The paper would benefit from clarifying which classes of ML problems most require spatial fairness analysis and where existing fairness frameworks may still be adequate.

I have the follwoing suggestions for improvement. 1). May be the authors shall provide one or two concrete case studies demonstrating how spatial aggregation or geographic proxies can distort fairness evaluation. 2). It will be great to include some spatial fairness evaluation protocol, including recommended reporting practices and diagnostic checks. 3). Provide Discussion on how spatial fairness interacts with existing fairness definitions such as demographic parity or equalized odds.

**Support:**

4

---

> ### Author Rebuttal · Authors · 2026-03-31
>
> We thank the reviewer for their constructive comments. Responses follow below.
>
> **Empirical illustration.**
> Please see response to Reviewer 561h.
>
> **Evaluation protocol.**
> We will add the following text to the paper, along with discussion of our recent relevant work.
>
> We propose a three-step audit protocol:
>
> - Spatial Proxy Analysis to identify where geographic features correlate with protected attributes;
>
> - Multi-Scale Auditing to ensure fairness metrics remain stable across different (reasonable) geographic boundaries (e.g., zip codes, census tracts) to defend against MAUP;
>
> - Dependency Clustering using statistics like Moran’s I to reveal localized disparities or redlining that global averages often mask. This ensures that fairness is a robust, verifiable property of the model rather than a coincidental artifact of how geographic data is aggregated.
>
> **Relationship with existing fairness metrics.** We will add the following:
>
> **Section 4.1.1. Limitations of existing fair-AI definitions/metrics.**
>
> Spatial structure does not replace existing fairness definitions, but rather introduces an additional layer that interacts with them. While classical fairness notions are defined over the outcome space, spatial dependencies introduce correlations across individuals or regions, which can lead to systematic disparities that are not captured by standard metrics. We examine how spatial data can undermine two broad families of fairness metrics in distinct ways.
>
> *Group fairness metrics.* Standard metrics like demographic parity or equalized odds evaluate outcomes across subgroups of a protected attribute (e.g., race: white/black). When location is treated as a protected attribute, issues arise because they weren’t designed for the messiness of spatial data. E.g, high cardinality (having thousands of zip codes as groups) makes reliable estimation of outcome rates per group difficult for metrics like demographic parity and calibration. MAUP also has an effect, e.g., model may satisfy demographic parity under one administrative partition (e.g., census tracts) but violate it under another (zip codes).
>
> Beyond boundaries, statistical assumptions of these metrics often fail in a spatial context. For example, equalized odds relies on the independence of data points, but Tobler’s First Law of Geography (nearby things are more related) means that observations are naturally correlated. Spatial autocorrelation violates the core statistical assumption of independence of tests used to evaluate bias.
>
> *Individual fairness metrics.* These metrics require similar individuals to receive similar outcomes, with similarity defined by a task-specific distance metric. Prior spatial fairness work has operationalized this by defining individual similarity as a function of the distance b/w two individuals’ locations, without accounting for other individual attributes. This formulation is legally insufficient since location is not a legally protected attribute in isolation. The true ethical concern arises when geography acts as a proxy for protected attributes like race.
>
> We propose that the path forward consists of joint metric frameworks. Instead of treating location and protected attributes independently, standard metrics can be extended by aggregating them over spatial neighborhoods, or by introducing spatial regularization terms that enforce fairness across similar regions. This would be consistent with the underlying legal & ethical concerns, while allowing practitioners to retain familiar fairness criteria & account for spatial effects that are otherwise overlooked.
>
>
> [21] Torino, G., 2024. Mestizo urbanism: Enduring racial intersections in Latin American cities. Journal of Latin American Studies, 56(1), pp.37-62.
>
> [22] Tan, S.B., 2023. Do ethnic integration policies also improve socio-economic integration? A study of residential segregation in Singapore. Urban Studies, 60(4), pp.696-717.
>
> [23] Jones, K., Manley, D., Johnston, R., & Owen, D. (2018). Modelling residential segregation as unevenness and clustering: A multilevel modelling approach incorporating spatial dependence and tackling the MAUP. Environment and Planning B: Urban Analytics and City Science, 45(6), 1122-1141.
>
> [24] Angwin, J., & Larson, J. (2017). California regulators require auto insurers to adjust rates. ProPublica. https://www.propublica.org/article/california-regulators-require-auto-insurers-to-adjust-rates
>
> [25] Sottek, T. C. (2016, May 8). Amazon working to address racial disparity in same-day delivery service. The Verge. https://www.theverge.com/2016/5/8/11634830/amazon-same-day-delivery-racial-bias-pledge
>
> [26] Raza, S., Shaban-Nejad, A., Dolatabadi, E. and Mamiya, H., 2024. Exploring bias and prediction metrics to characterise the fairness of machine learning for equity-centered public health decision-making. IEEE Access, 12, pp.180815-180829.

---

> > ### Author Rebuttal · Reviewer_XP7r · 2026-04-01
> >
> > Thanks for the responses. We will keep our positive scores.

---

### Official Review · Reviewer_561h · 2026-03-10

**Significance:** 4
**Argument Clarity:** 3
**Rating:** 5
**Confidence:** 4

**Questions:**

- The paper discusses the shortcomings in related work for current spatial fair literature. How to fix them? Are there technical problems or intrinsic difficulties in algorithmic design?

**Alternative Views Section:**

Yes

**Compliance With Llm Reviewing Policy A Conservative:**

Affirmed.

**Discussion Potential:**

3

**Paper Summary:**

This paper argues for the importance of spatial fairness. The authors delineate the need to address persisting bias in spatial decision-making contexts and legal efforts proposed to address it, and outline problems unique to spatial analysis that compound the difficulties in addressing spatial bias computationally. Moreover, they identify drawbacks of the handful of spatial fairness work proposed so far to highlight why they fall short. The paper also establishes a set of guidelines to support future spatial fairness research avoid similar pitfalls
and achieve maximal impact with adoption in the real world.

**Position:**

Yes

**Position In Title:**

No

**Related Work:**

4

**Strengths And Weaknesses:**

Strengths:
- The topic of spatial fairness is novel and relevant.
- The spatial fairness is a real need. Though it suffers from technical difficulties, it is essential to study how to reduce bias in spatial information.

Weaknesses:
- The paper does not provide concrete examples of the outcome of spatial bias and how to reduce it.

**Support:**

3

---

> ### Author Rebuttal · Authors · 2026-03-31
>
> We thank the reviewer for their constructive comments. While we included illustrative examples of detrimental outcomes of spatial bias, we did not provide a full treatment of a real-world example of spatial bias leading to discriminatory outcomes & attempts to address it. We will add the following.
>
> **Addition of empirical illustration and concrete examples of spatial bias and attempts to address it:**
>
> Conceptually, MAUP can be considered a spatial analogue of Simpson’s Paradox. While the latter shows how statistical trends can disappear or even reverse based on how data is partitioned into subgroups, MAUP demonstrates that these same reversals can occur based on how geography is partitioned. As Fotheringham and Wong (1991) showed, simply changing the scale or boundaries of geographic units can cause statistical relationships to reverse sign or lose significance entirely. This sensitivity is particularly acute in multivariate analyses, which are common in fair-AI work. This fragility is well documented across disciplines, e.g., Jones et al. [23] show that the same data can yield conflicting conclusions about residential segregation depending on the spatial unit chosen. MAUP also affects ecological model inferences in disease mapping due to substantial variation in both scale and zoning effects [27].
>
> Concrete, real-world examples illustrate this distortion and the tangible harms of ignoring the correlation between location and protected attributes. For instance, a ProPublica investigation found that auto insurers charged drivers in minority neighborhoods premiums up to 30% higher than drivers of equivalent risk residing in predominantly white areas (Angwin et al., 2017). This discovery prompted California regulators to mandate corrective measures [24]. Similarly, Bloomberg revealed that Amazon Prime’s same-day delivery service disproportionately excluded majority-Black neighborhoods, a disparity that income differences could not justify. After pressure from members of Congress and the mayors of New York City and Boston, Amazon committed to fill all racial gaps in the service’s coverage [25]. In both instances, the bias was spatial: individuals were treated differently based on where they lived because geography acted as a proxy for race, and identifying this spatial pattern was the essential first step towards corrective action and regulatory accountability. Another real-world example of MAUP is gerrymandering: a fixed population’s representation can be manipulated simply by redrawing boundaries. In a fairness context, a county-level audit may show no racial disparity in mortgage denials, while a census-tract-level analysis of the same data may reveal a different conclusion.
>
> **How to fix shortcomings in related work**
>
> As the reviewer points out, we discuss both the challenges unique to spatial data that complicate spatial fairness (Section 3) and the shortcomings of the current spatial fairness literature (Section 4). Our goal in this paper, however, is to articulate and structure these open problems rather than to propose full technical solutions for each of them. We view that as beyond the scope of a position paper, since each challenge, both conceptual and algorithmic, would likely require its own dedicated methodological study. That said, Section 5 outlines research guidelines and directions, and we will revise the paper to make clearer how each guideline connects back to the specific limitations identified in Sections 3 & 4.
>
> [5] Wang, F., 2020. Why public health needs GIS: a methodological overview. Annals of GIS, 26(1), pp.1-12.
>
> [6] Knaap (2017). The cartography of opportunity: Spatial data science for equitable urban policy. Housing Policy Debate, 27(6), 913-940.
>
> [8] Majumdar et al. (2022) from paper.
> [9] Lee et al. (2025). The modifiable areal unit problem in political science. Political Analysis, 33(4), 412-424.
>
> [10] Manley (2021). Scale, aggregation, and the modifiable areal unit problem. In Handbook of regional science (pp. 1711-1725).
>
> [11] Sacharidis et al. (2023) from paper.
> [12] Ratliff et al. (2025). Don’t trust a single gerrymandering metric. La Matematica, 4(3), 764-809.
> [15] Barocas et al. (2016) from paper.
>
> [16] Chen et al. (2021). Ethical machine learning in healthcare. Annual review of biomedical data science, 4(1), 123-144.
>
> [17] Rodolfa et al. (2021). Empirical observation of negligible fairness–accuracy trade-offs in machine learning for public policy. Nature Machine Intelligence, 3(10), 896-904.
>
> [18] Musterd et al (2017). Socioeconomic segregation in European capital cities. Increasing separation between poor and rich. Urban geography, 38(7), 1062-1083.
> [19] Weerts et al. (2023). Algorithmic unfairness through the lens of EU non-discrimination law: Or why the law is not a decision tree. ACM FAccT (pp. 805-816).
>
> [20] Meding  (2025). It's complicated. The relationship of algorithmic fairness and non-discrimination regulations in the EU AI Act. arXiv e-prints, arXiv-2501.

---

> > ### Author Rebuttal · Reviewer_561h · 2026-04-02
> >
> > Thank you for the reply.

---

### Official Review · Reviewer_puVj · 2026-03-13

**Significance:** 3
**Argument Clarity:** 3
**Rating:** 5
**Confidence:** 2

**Questions:**

Do the authors have a relevant experience implementing spatially-fair AI systems? If yes, how does it correlate with the challenges outlined in the work?

**Alternative Views Section:**

Yes

**Compliance With Llm Reviewing Policy A Conservative:**

Affirmed.

**Discussion Potential:**

2

**Final Justification:**

My concerns were addressed during the rebuttal.

**Paper Summary:**

This work focuses on the "spatial fairness" problem, which arises from implicit discrimination based on geospatial proxies (e.g., ZIP codes used as a proxy for neighborhoods dominated by a particular race). It provides an overview of existing challenges, problems, and solutions, and proposes a set of guidelines to better address spatial unfairness.

**Position:**

Yes

**Position In Title:**

No

**Related Work:**

3

**Strengths And Weaknesses:**

**Strengths:**

- The paper is well-written and easy to follow. It presents each claim clearly.
- The evidence base is solid and rather rich: various works on specific issues, such as discrimination in the mortgage industry or by delivery companies, limitations of "fairness through unawareness", and discriminative patterns in house pricing, are leveraged to support the claims.
- A thorough overview of the challenges that are specific to spatial data is included in the paper.

**Weaknesses:**

1. From my understanding of the Position Paper Track's guidelines, **this paper does not state the position in the title**.

   I suggest the following improvements:
   - The title can be changed to something that explicitly advocates for putting more effort into the spatial fairness research. For example,

      **Position: We Are Still Far from Achieving Spatial Fairness in AI**
   - The corresponding position should be clearly stated in the beggining of the main text (for example, as clear as in the abstract, lines 018-020).
1. The "Alternative Views" section is rather brief and lacks references to works that support the listed alternatives. This hinders a well-rounded scientific discussion of the work. Since the discussion potential is one of the key metrics for evaluating position papers, I kindly suggest the authors to expand this section and provide concrete references to the mentioned alternative views.
1. Connections to US legislation limit the immediate applicability of some of the guidelines to the rest of the world. It even comes to entire subsections being built upon concepts that might be (legally) specific to the US (for example, Section 5.2.2, which is inspired by a very specific treatment of immutable characteristics in the US). However, to be fair, the authors acknowledge this limitation in the text. Perhaps additional recommendations should be given in cases when a certain guideline might not be widely applicable.

**Minor issues:**

1. I respectfully suggest formatting tables with the `booktabs` package, as recommended in the template.
1. The subsubsection style is not consistent with the rest of the template.
1. Line 289 col. 2 - a space after the period is missing.

**Support:**

3

---

> ### Author Rebuttal · Authors · 2026-03-31
>
> Thank you for the constructive comments and pointing out our mistake re: the title.
>
> We will change the title to:
>
> *Position: Why Current Fair-AI Fails Spatial Fairness, And How to Adapt to It*
>
> **Please see other responses below.**
>
> **Alternative views.**
> Per the reviewer’s suggestion, we will add these specific citations to Alternative Views:
>
> **Alternative View #1.**
> While spatial inequities often originate from historical policy decisions, policymakers already rely on computational methods to address them. For example, research in public policy [2,6], public health [5, 26], and urban planning [1,3,4] already deploys GIS, spatial statistics, and ML to study [1–4, 26] and mitigate them [5,6]. As the public policy community already uses them in addressing spatial inequities in various domains, excluding computational methods from the conversation would be a false dichotomy.
>
> **Alternative View #2.**
> Traditional fair-AI is insufficient because:
> (1) Spatial autocorrelation violates the independence (i.i.d.) assumption on which most such approaches rely. Research shows ignoring spatial autocorrelation when applying fair-AI methods actively distorts bias detection [8], which can cause these methods to produce unreliable fairness assessments.
> (2) Fair-AI metrics are brittle to MAUP. Research shows conducting the same analysis with data aggregated to different spatial units can yield very different conclusions [9,10], and how existing work is not robust to MAUP[11]. A real world example, gerrymandering, shows how tricky it can be to deal with it successfully [12].
>
> **Alternative View #3.**
> Optimizing for accuracy or operational efficiency will often lead to lower fairness and vice versa. Yet, this has not prevented fair-AI approaches since:
> (1) Many domains have a legal mandate to mitigate bias against sensitive groups regardless of the resulting cost to accuracy or efficiency [15].
> (2) Even in high-stakes domains with no mandates, such as healthcare or criminal justice, there is broad ethical consensus that equitable outcomes should be prioritized over marginal accuracy or gains in system performance [16].
> (3) Empirical evidence suggests this trade-off is often smaller than expected: well-designed fairness interventions can substantially reduce disparities with negligible loss in accuracy [17].
> These arguments apply equally to spatial fairness.
>
>
> **Focus on U.S. law.**
>
> Despite a focus on US law, we argue our framework generalizes beyond the U.S. for 3 reasons.
>
> - Spatial inequity is a global challenge. Correlation b/w location and protected characteristics is well-documented across Europe, Latin America [21], Asia [22], and beyond [18].
>
> - While legal terminology varies, the underlying rationale translates across jurisdictions. E.g., EU’s GDPR and AI Act establish non-discrimination obligations in algorithmic systems in high-risk systems that functionally overlap with our guidelines’ goals [19, 20] albeit via different legal concepts (e.g., indirect discrimination) under EU equality directives [19]. Similarly, Brazil’s Estatuto da Cidade (Law 10.257/2001) requires urban policies to mitigate socio-spatial segregation, which overwhelmingly affects the country’s Afro-Brazilian population, and to ensure equitable distribution of the benefits and burdens of urbanization.
>
> - Ethical justifications for these guidelines, i.e., protecting marginalized communities from location-based harm, stands independently of specific statutory protections.
>
> To make this clearer, we will (1) reference equivalent non-U.S. frameworks (e.g., EU equality directives), and (2) explicitly state the universal ethical justifications that apply where legal frameworks vary.
>
> **Relevant experience.**
>
> Yes, our prior work directly informs this work. We showed that existing work lacks robustness to MAUP, as well as treating location alone as a protected attribute can lead past work to incorrectly flag unbiased decisions as discriminatory. This occurs because location often carries legitimate, decision-relevant information that is legal to consider (e.g., high-fire-risk zones). The core issue is location's correlation with protected attributes like race. We have shown that ignoring these valid spatial risks lead to unreliable fairness assessments.
>
> We limited self-citations to preserve anonymity. If accepted, we will add a full discussion of our findings.
>
>
> [1] Aaronson et al. (2021). The effects of the 1930s HOLC “redlining” maps. American Economic Journal: Economic Policy, 13(4), 355-392.
>
> [2] Meier  et al. (2022). Tracing the legacy of redlining: a new method for tracking the origins of housing segregation. National Community Reinvestment Coalition.
>
> [3] Moro et al. (2021). Mobility patterns are associated with experienced income segregation in large US cities. Nature communications, 12(1), 4633.
>
> [4] Xu et al. (2025). Using human mobility data to quantify experienced urban inequalities. Nature human behaviour, 9(4), 654-664.

---

> > ### Author Rebuttal · Reviewer_puVj · 2026-04-05
> >
> > I thank the authors for their thorough response! I particularly appreciate the inclusion of additional alternative views.
> >
> > With these changes implemented, I believe this is a solid work. **I therefore raise my score to "Accept".**
> >
> > The only reason I cannot assign a higher score is the continued strong reliance on U.S. legislation (even with the proposed amendments in place). If the authors find the time and energy to support each of their legislative points with more universal references (such as a comprehensive survey of laws across multiple jurisdictions), that would be excellent.

---

### Decision · Program_Chairs · 2026-04-30

**Decision:**

Accept (regular)

**Comment:**

The paper received three positive reviews (three accepts). There was general appreciation for the position taken by the paper around spatial fairness backed by strong evidence, clear motivation, and thorough discussion cutting across disciplines (ML, public policy). There were a few concerns raised, most of which were adequately addressed during the rebuttal phase. Consequently, an accept consensus was reached.